# Long-Time Coherent Integration for Marine Targets Based on Segmented Compensation

**Zhenfang Zhao** [1] , **Yisong Zhang** [1,2] **, Wenguang Wang** [1,*] **, Ben Liu** [3] **and Wei Wu** [4]

1    School of Electronic and Information Engineering, Beihang University, Beijing 100191, China; zhaozhenfang_0509@buaa.edu.cn (Z.Z.); gsjrzys_hq@bank-of-china.com (Y.Z.)
2    Bank of China, Beijing 100032, China
3    Institute of Deep-Sea Science and Engineering, Chinese Academy of Sciences, Sanya 572000, China; liub@idsse.ac.cn
4    College of Weaponry Engineering, Naval University of Engineering, Wuhan 430030, China; 1712021043@nue.edu.cn
*    Correspondence: wwenguang@buaa.edu.cn; Tel.: +86-10-8231-7240

**Abstract:** Long-time coherent integration is an effective method for dim target detection from heavy sea clutter. To detect dim targets, a novel long-time coherent integration method based on segmented compensation is proposed in this paper. The method models the complex motion of a marine target as the combination of multi-stage uniformly accelerated motions. According to the difference of energy distribution in Doppler frequency domain, this method can suppress sea clutter and detect the regions of interest (ROIs). Using time–frequency domain energy analysis, the potential target can be extracted. After estimating the parameters and segmentation, for the potential targets, the phase compensation factor can be used to eliminate the Doppler frequency modulation caused by the complex motion. Finally, for the compensated signal, long-time coherent integration is performed to realize the target detection and discrimination under low signal-to-clutter ratio. To verify the effectiveness of the proposed method, we apply simulation data and measured CSIR data in the experiments. The results show that the proposed method can integrate the energy of target more effectively than MTD and RFrFT, and the novel method has better detection performance for complex moving targets under low signal-to-clutter ratio situation.

**Keywords:** parameter estimation; motion segmentation; phase compensation; coherent integration

## 1. Introduction

Target detection in the background of sea clutter is always an important problem in radar applications. Sea clutter, affected by many factors such as grazing angle, wind, polarization, and so on, has a very large dynamic range. Both the low signal-to-clutter ratio (SCR) caused by the heavy sea clutter and the complex motion of the marine targets are the factors reducing the detection performance. In addition to being affected by their own power, the motion of small marine targets is also affected by other factors such as sea waves and wind blowing, resulting in strong time variability. The stable detection of maneuvering marine targets has always been a challenging problem in water traffic surveillance, anti-smuggling, maritime search and rescue, and so on.

Long-time coherent integration is an important method to effectively increase the SCR of dim targets to improve the detection probability [1]. Long-time integration is mainly divided into two types according to the coherence of the signal, namely incoherent integration and coherent integration. Typical incoherent integration includes Hough transform [2,3], Radon transform [4], tracking before detection [5,6], and so on. Incoherent integration is just the superposition of echo in amplitude without considering the phase of signal, so the integration performance is not so good at low SCR. Compared with incoherent integration, coherent integration takes into account both the amplitude and

phase, and can obtain higher SCR than that of incoherent integration. During the long observation, the envelope of the target echo will migrate across different range bins. Moreover, the complex motion of the target will cause the Doppler frequency spectrum expansion. So, the correction of the range migration (RM) and the compensation of the Doppler frequency are the main problems in the long-time coherent integration. Keystone transform (KT) is a range migration correction method widely used in synthetic aperture radar imaging. The core of KT is to scale the time axis of the echo in the range–frequency domain [7] so as to correct the linear range migration caused by the target motion [8,9]. But KT is not effective for complex movement because it cannot correct the range curving caused by the high-order movement of the target, nor can it eliminate the Doppler ambiguity [10]. Ref. [11] proposed an algorithm which applied azimuth dechirping and the second-order keystone transform to eliminate the Doppler ambiguity to correct the range curvature. RFT [12] is another method widely used for target range migration correction. It can perform coherent integration along a model-based track on the range–time plane [13], which can correct the range curvature caused by the higher-order motion of the target. However, it depends on the search of the motion parameters of different models, and the calculation complexity is relatively high [14]. To compensate the Doppler frequency modulation (DFM) caused by the high-order motion of the target, many methods have been proposed successively to improve the integration effect of the target echo, such as the time reversal transform [15,16], fractional Fourier transform (FrFT) [17,18], Lv's distribution (LVD) [19], etc. Among them, fractional Fourier transform eliminates Doppler frequency modulation through the rotation of the time–frequency plane, which needs to search the acceleration of the moving target [20]. LVD gives the natural representation of LFM signal in the center frequency-chirp rate (CFCR) domain [21], which transforms the signal from the time–frequency domain to the CFCR domain by scaling the parametric symmetric instantaneous autocorrelation function (PSIAF). Ref. [22] proposed a fast non-searching method based on the adjacent cross-correlation function (ACCF) and LVD, which can remove the range migration and reduce the order of Doppler frequency modulation to realize the coherent integration, target detection and parameter estimation for maneuvering target. The method in Ref. [23] is based on frequency spectrum segment processing (FSSP) and the segmental Lv's distribution (SLVD), where the FSSP divides the received signal into several subband signals and expands the range resolution of the subband signals to eliminate the RM and SLVD is applied to achieve the coherent integration of the subband signals. By combining Radon transform with FrFT, RFrFT can compensate Doppler frequency modulation while correcting the range migration [24]. To achieve long-time coherent integration of uniformly accelerated targets, the method in Ref. [25] extracts signals along the track on the range–time plane through a multidimensional parameter search, and converts the correction of range migration and the phase compensation into the matching of motion parameters. Based on RFrFT, W. Wang proposed a two-stage RFrFT long-time coherent integration for marine targets [26] to address the problem of excessive computation caused by multidimensional parameters search. The method includes two steps: the first step detects regions of interest (ROIs) based on the spectral similarity between adjacent range bins, and then performs out RFrFT integration and target discrimination within ROI regions. It reduces the calculation cost of RFrFT and has good detection performance at the same time. However, due to the diversity and complexity of the influencing factors, the marine target always has a complex motion, which is difficult to accurately describe during the long-time integration by using a single uniformly accelerated model. As a result, it will be difficult to completely eliminate the problem of Doppler frequency modulation caused by the target motion. Aiming at the complex motions of marine targets, this paper proposes a novel long-time coherent integration method based on segmented phase compensation. On the basis of extracting potential targets with low threshold, the motion of the targets is decomposed into a combination of multi-stage uniformly accelerated motions. After estimating the motion parameters of potential targets using the symmetric instantaneous autocorrelation function (SIAF), the phase

compensation is performed to eliminate Doppler frequency modulation for each segment. Finally, the complex moving targets can be detected using long-time coherent integration.

The contributions of this paper are as follows:

1.  Aiming at the problem of mismatch between the complex motion and the single motion model, this paper presents a new modeling method that decomposes the complex motion of the target into the combination of multiple uniformly accelerated motions to achieve a simplified description.
2.  For each segment, the parameters under low SCR are estimated under the model constraints, and then the compensation factor is constructed according to the parameter estimation to compensate the secondary order phase to eliminate the Doppler frequency modulation caused by the complex motion.
3.  To eliminate the false alarms that may exist in the detection results, a target discrimination method based on the 3 dB spectrum width of the symmetric instantaneous autocorrelation function is proposed, which can effectively distinguish the false alarm caused by the sea clutter.

The structure of this paper is as follows: In Section 2, the echo model of moving target and some classic coherent integration methods are introduced. In Section 3, a long-time coherent integration method based on segment compensation is presented, including detection of ROIs, parameter estimation and motion segmentation of potential targets, long-time coherent integration and target discrimination. In Section 4, the effectiveness of the proposed method is verified based on simulation data and measured data. Finally, the work of this paper is summarized and some useful conclusions are given.

## 2. Signal Processing Models

### 2.1. Marine Target Echo Model

In order to obtain higher range resolution, linear frequency modulation (LFM) is widely used as radar transmitting signal; its form can be expressed as Equation (1).

$$s(t) = rect(\frac{t}{T_p}) \exp[j2\pi(f_c t + \frac{kt^2}{2})] \tag{1}$$

where $rect(u) = \begin{cases} 1, |u| \le \frac{1}{2} \\ 0, |u| > \frac{1}{2} \end{cases}$, $f_c$ is the center frequency, $k = B/T_p$ is the chirp rate of the LFM signal whose bandwidth is $B$ and pulse width is $T_p$.

The motion of the marine target will cause the radial distance between the target and the radar to change at any time. If the distance between the target and the radar at the time $t_m$ is $R(t_m)$, the echo delay will be $\tau(t_m) = 2R(t_m)/C$, where $C$ denotes the speed of light and the received echo of the radar can be expressed as Equation (2).

$$s_r(t, t_m) = \sigma_r rect(\frac{t - \tau(t_m)}{T_p}) \exp[-j2\pi f_c \tau(t_m)] \exp\{j\pi k[t - \tau(t_m)]^2\} \tag{2}$$

where $\sigma_r$ is the complex scattering coefficient of the target. After pulse compression, the form of the signal is as Equation (3).

$$s_{pc}(t, t_m) = A_0 \sin c\{B[t - \tau(t_m)]\} \exp[-j2\pi f_c \tau(t_m)] \tag{3}$$

In Equation (3), $A_0$ is related to $\sigma_r$, which can be approximated as a constant. If the changing of the range in long-time integration exceeds the range resolution of the radar, the motion of the target will cause migration of the echo envelope, and the integration based on a single range bin will not be very effective.

The motion of the marine target is complex and time-varying, but the acceleration of the target is relatively stable in a short time, which means the target motion can still be regarded as the uniform acceleration movement in a short time. For the marine target,

if the initial radial velocity is $v_0$, the acceleration is $a$, then the instantaneous velocity $v(t) = v_0 + at$, and the instantaneous Doppler frequency $f_d(t) = 2(v_0 + at)/\lambda$, so the second-order phase caused by the acceleration can be expressed as Equation (4).

$$\phi(t) = 2\pi \int_0^t f_d dt = \frac{4\pi}{\lambda}(v_0 t + \frac{1}{2}at^2) \tag{4}$$

To simplify the expression, the motion of marine targets can be modeled as a combination of multi-stage uniformly accelerated motions. The radial distance corresponding to the $i$th segment can be expressed as Equation (5).

$$R_i(t_m) = R_{i-1} + v_{i-1}(t_m - \sum_{j=0}^{i-1} t_j) + \frac{1}{2}a_i(t_m - \sum_{j=0}^{i-1} t_j)^2 \tag{5}$$

In Equation (5), $v_{i-1}$ is the radial initial velocity of the $i$th segment, $a_i$ is the acceleration, and the duration of the $j$th segment is $t_j$ and $t_0 = 0$, then $\sum_{j=0}^{i-1} t_j$ is the total duration before the $i$th segment.

Due to the non-uniform motion of the target, its Doppler frequency changes with time, which will cause the problem of Doppler frequency modulation. As a result, the Doppler spectrum will disperse in a wide range of Doppler domains. The Doppler frequency of the echo generated by the movement of the $i$th segment is shown in Equation (6), and the Doppler spectrum width is proportional to $a_i$.

$$f_{di}(t_m) = \frac{2v_{i-1} + 2a_i t_m}{\lambda} \tag{6}$$

The echo of the $i$th segment can be compressed as Equation (7).

$$s_{pci}(t, t_m) = A_0 \sin c\{B[t - \frac{2R_i(t_m)}{C}]\} \exp[-j\frac{4\pi}{\lambda}R_i(t_m)] \tag{7}$$

If the complex motion of the target can be segmented into a combination of $N$ segments of uniformly accelerated motion, then the compressed echo of the marine target can be expressed as Equation (8).

$$s_{pc}(t, t_m) = \sum_{i=1}^N s_{pci}(t, t_m) \tag{8}$$

### 2.2. Coherent Integration

MTD is a widely used target detection method; the core of MTD is to improve the SCR of moving targets by using coherent integration, so it is also regarded as a set of Doppler filter banks [27], which can be defined as Equation (9).

$$X_{MTD} = \int s(t, r)e^{-j2\pi ft} dt \tag{9}$$

To perform MTD, the key processing is the Fourier transform (FT) to the echo in the same range bin [28]. Owing to the RM and DFM, the integration result is influenced by the motion of the target. In addition, the MTD is just a type of short-time integration procedure.

To solve the coupling between the range walking and slow time, RFT is proposed to realize long-time coherent integration. Different from the existing Radon transformation or FT, the RFT may be quite helpful for the detection of moving targets due to the correction of range migration [29]. The standard RFT combining the Fourier transform and the Radon transform is defined as Equation (10).

$$X_{RFT} = \int s(t, r_0 - v_0 t)e^{-j2\pi ft} dt \tag{10}$$

where $s(t, r_0 - v_0 t)$ is a two-dimension complex function defined in $(t, r)$ plane, and $r = r_0 - v_0 t$. Therefore, Equation (9) is a special case of Equation (10) when the target is very slow and there is no range migration. When the target moves uniformly and crosses the range gates, neither MTD nor RFT can integrate the target energy effectively. As a generalization of the FT, FrFT [30] is proposed for signal processing, which is defined as Equation (11).

$$X_\alpha(u) = \int x(t) K_\alpha(t, u) dt \qquad (11)$$

where $\alpha$ is the transformation angle; the value range of $\alpha$ is 0 to $\pi$, which can be calculated by $\alpha = p\pi/2$. $p$ is the transformation order, which can be determined by searching the acceleration of the moving target after dimensional normalization. The relationship between the acceleration and transformation order is shown in Equation (12).

$$p = -\frac{2 \operatorname{arccot}(2 a_i S^2 \lambda)}{\pi} + 2 \qquad (12)$$

In (12), $S$ is the scale factor, depending on the sampling frequency $f_S$ and integral time $T_n$, $S = \sqrt{T_n/f_S}$. The transform kernel $K_\alpha$ is defined as follows:

$$K_\alpha(t, u) = \begin{cases} A e^{(j\frac{u^2+t^2}{2}\cot\alpha - jut\csc\alpha)}, \alpha \neq n\pi \\ \delta(t - u), \alpha = 2n\pi \\ \delta(t + u), \alpha = (2n \pm 1)\pi \end{cases} \qquad (13)$$

where $A = \sqrt{(1 - j\cot\alpha)/2\pi}$, FrFT can be seen as a rotation of the time–frequency plane at an angle of $\alpha$. When $\alpha = \pi/2$, FrFT is the Fourier transform. Compared with Fourier transform, FrFT can compensate Doppler frequency modulation caused by the uniformly accelerated motion effectively, and is widely used for the long-time coherent integration of dim marine targets.

By combining FrFT and RFT, RFrFT [31] is proposed to achieve long-time coherent integration of maneuvering targets. RFrFT is defined as Equation (14).

$$X_{RFrFT} = \int s(t, r_0 - v_0 t - \frac{1}{2}at^2) K_\alpha(t, u) dt \qquad (14)$$

where the range $r = r_0 - v_0 t - \frac{1}{2}at^2$ represents the accelerated or high-order motion of the target, which is used to search the echo of moving target in the plane $(t, r)$.

RFrFT can extract the echoes of the target on the range–time plane according to the motion model. In order to detect dim targets from noise and heavy sea clutter, RFrFT must search for different parameter combinations of $[r_0, v_0, a]$; this results in high computation. RFrFT can be regarded as a kind of extension of the MTD, FrFT and standard RFT, which can be used effectively in target detection from very low SCNR and even from the situation with range migration and Doppler frequency modulation.

## 3. Long-Time Coherent Integration Based on Segmented Compensation

Based on the models in Section 2, to the complex moving target, such as the variable acceleration moving, a long-time coherent integration method based on segmented compensation is proposed in this section. The proposed long-time coherent integration includes the processing of ROI detection, motion estimation and segmentation, phase compensation and coherent integration, and target discrimination. Below is the framework of the proposed method, which is shown in Figure 1. To obtain the potential targets, firstly, the ROIs are detected in the range–Doppler domain. Then, the velocity and acceleration of each ROI is estimated to realize the motion segmentation. Based on the parameters estimation and motion segmentation, the phase factor is constructed to eliminate Doppler frequency modulation caused by the movement of the target. For the compensated signal,

long-time coherent integration is performed and target detection is realized. After target discrimination, the false alarms caused by sea clutter can be eliminated.

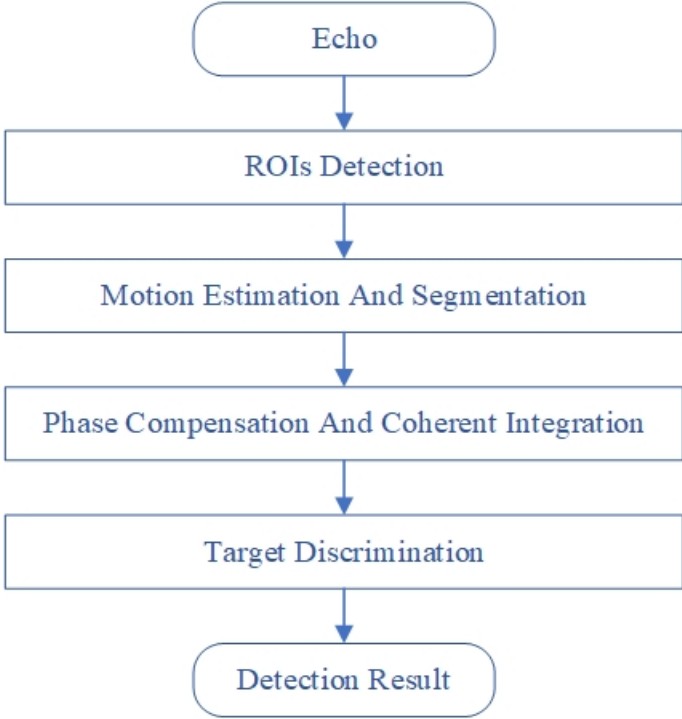

**Figure 1.** Framework of long-time coherent integration based on segmented compensation.

### 3.1. ROI Detection

This section includes three modules: wavelet packet decomposition, ROI detection and potential target signal extraction.

In radar surveillance of the sea, sea clutter is the main component of the echo; small marine targets are often submerged in them. In radar echoes, the energy of sea clutter is distributed in all range bins, while the target only appears in specific range bins. Because the motion of the target is subject to specific constraints, its Doppler spectrum characteristics are different from those of sea clutter. The difference in energy distribution between the target and the sea clutter can be used to distinguish the target from the sea clutter.

Wavelet packet decomposition (WPD) can decompose the signal into the components with different frequencies, which can characterize the energy distribution. WPD can decompose the signal into two parts using a set of orthogonal wavelet bases: high frequency and low frequency, and then each part is used as input, and will be decomposed again. In this way, the corresponding components $c_j (j = 1, 2, \ldots, 2^i)$ with different frequency bands and satisfied $\sum_{j=1}^{2^i} c_j(n) = s(n)$ can be obtained at the *ith* layer.

To each range bin, the WPD provides the energy distribution with frequency. And the energy in the *j*th frequency band of the *mth* range bin can be expressed as Equation (15).

$$E_{m,j} = \sum_n [c_{m,j}(n)]^2 \tag{15}$$

In (15), $m = 1, 2, \ldots, M$ is the index number of the range bin. After extracting the energy distribution, the echo is decomposed into $M \cdot 2^i$ frequency units $P_{m,j}$ in the frequency domain, and the energy of each frequency unit is $E_{m,j}$.

The high-energy units can be detected by comparison with the threshold $T_m$, defined as Equation (16).

$$T_m = mean(E_{m,j}) + \eta_m \cdot std(E_{m,j}) \tag{16}$$

where $\eta_m$ is the threshold coefficient, then a frequency unit can be judged whether it is a high-energy unit.

After the high-energy unit detection, the higher unit numbers in all range bins will be used to analyze whether a frequency unit belongs to sea clutter or a potential target. If most range bins are judged to be high-energy units, a frequency unit will be considered to belong to sea clutter, because it is impossible for a marine target to pass through a lot of range bins and show high energy in many range bins in a short time. The sea clutter can be partly suppressed by using the high-energy unit numbers to weight each frequency unit, such as Equation (17).

$$E_{m,j}{}' = (1 - \frac{M'}{M})E_{m,j} \tag{17}$$

where $M'$ is the unit numbers that are judged as high-energy in the $M$ range bins. To the sea clutter, its $M'$ is close to $M$, but to the target, $M'$ is much smaller than $M$. This difference makes the potential targets in the scene can be extracted. The region where the potential target is located is called the region of interest (ROI).

Let the extracted $K$th ROI be $ROI_k(k = 1, 2, \ldots, K)$ with the range $R_k$, and the Doppler frequency $D_k$. If the $R_k$ includes multiple adjacent range bins, we should select the the right bin to provide echo. To solve this problem, the signals in $R_k$ are divided into $L$ segments in time domain. To each segment, the signal is transformed into the Doppler frequency domain, then the bin with the highest energy of each segment in $D_k$ will be extracted. The extracted segments are connected successively to complete the signal extraction of potential targets.

### 3.2. Motion Estimation and Segmentation

In motion estimation and segmentation, the initial velocity of the signal in each short period is estimated, and the acceleration between segments is estimated according to the initial velocity between adjacent segments. Finally, the motion is segmented according to the estimated parameters.

The motion of the marine target is not only controlled by its own power, but also affected by environmental factors such as sea waves and wind. In a long-time observation, the variation of the influencing factors leads to the complex motion of the marine target, which is difficult to describe using a certain model. To model the complex movements, the motion can be decomposed into multiple fragments. In each short period, the target acceleration can be considered as relatively stable and can be approximated to a constant value, so the complex motion of the marine target can be decomposed into several short-time uniformly accelerated motions. The parameters of the uniformly accelerated motion of each stage are estimated respectively, and the phase compensation factor is constructed to eliminate the Doppler frequency modulation caused by the uniformly accelerating motion of the target.

In the extracted ROIs, the movement of the $i$th segment can be simplified as Equation (18) in the slow time dimension.

$$s_i(t_m) = A_0 \exp[j2\pi f_{di}(t_m - \sum_{j=0}^{i-1} t_j) + j\pi k_i{}'(t_m - \sum_{j=0}^{i-1} t_j)^2] \tag{18}$$

where $f_{di} = \frac{2v_{i-1}}{\lambda}$ is the Doppler center frequency of the signal in the $i$th segment, which can be calculated from the initial velocity in the $i$th segment; $k_i{}' = \frac{2a_i}{\lambda}$ is the Doppler modulation rate of the signal in this segment; and $a_i$ is the acceleration in the segment.

Then, the symmetric instantaneous autocorrelation function (SIAF) [32] of $s_i(t_m)$ at time $t_{0i}$ can be obtained as Equation (19).

$$f_i(t_{0i}, \tau) = s_i(t_{0i} + \frac{\tau}{2})s_i{}^*(t_{0i} - \frac{\tau}{2}) = A_0{}^2 \exp[j2\pi f_{di}\tau + j2\pi k_i{}'(t_{0i} - \sum_{j=0}^{i-1} t_j)\tau] \tag{19}$$

where $(\cdot)^*$ denotes the complex conjugation, for a given $t_0$; $f_i(t_0, \tau)$ is a one-dimensional function $f_i(\tau)$, its Fourier transform has the form as Equation (20).

$$F_i(f_\tau) = A_0{}^2 \sin c\{f_\tau - [f_{di} + k_i{}'(t_{0i} - \sum_{j=0}^{i-1} t_j)]\} \approx A_0{}^2 \sin c(f_\tau - f_{di}) \qquad (20)$$

Due to the short duration of each motion segment and the limited small acceleration of the marine target, the item $k_i{}'(t_{0i} - \sum_{j=0}^{i-1} t_j)$ can be negligible to the target initial velocity estimation. The Doppler frequency $f_{\max i} = \arg\max_{f_\tau} [F_i(f_\tau)]$ corresponds to the peak of the transformed spectrum. Then, the estimated Doppler frequency $\hat{f}_{di}$ for the motion in the $i$th segment can be calculated as Equation (21).

$$\hat{f}_{di} = \frac{\sum\limits_{f_j = f_{\max i} - \Delta f}^{f_{\max i} + \Delta f} f_j \cdot F_i(f_j)}{\sum\limits_{f_j = f_{\max i} - \Delta f}^{f_{\max i} + \Delta f} F_i(f_j)} \qquad (21)$$

where $\Delta f$ is the Doppler window width, then the initial velocity of the $i$th segment can be estimated by $\hat{v}_{i-1} = \frac{\hat{f}_{di}\lambda}{2}$. The acceleration estimation, as shown in Equation (22), can be obtained from the velocity of two adjacent segments.

$$\hat{a}_i = \frac{\hat{v}_i - \hat{v}_{i-1}}{t_{0i} - t_{0(i-1)}} \qquad (22)$$

### 3.3. Phase Compensation and Long-Time Coherent Integration

According to the estimated motion parameters, the phase compensation factor can be constructed. The core of phase compensation is to eliminate the quadratic phase term caused by the uniform acceleration motion of the target. For the $i$th segment, the phase compensation factor is $\phi_i$, which has the form of Equation (23). After the phase compensation, the coherent integration is performed to improve the SCR of the echo.

$$\phi_i(t_m) = \frac{1}{\lambda}[2(\hat{v}_{i-1} - \hat{v}_0)(t_m - \sum_{j=0}^{i-1} t_j) + \hat{a}_i(t_m - \sum_{j=0}^{i-1} t_j)^2] \qquad (23)$$

After phase compensation, the phase of the signal in $i$th segment will not contain $t^2$ term anymore, in the form of Equation (24).

$$s_i{}'(t_m) = s_i(t_m) \cdot \exp[-j2\pi\phi_i(t_m)] = A_0 \exp[j\frac{4\pi}{\lambda}v_0(t_m - \sum_{j=0}^{i-1} t_j)] \qquad (24)$$

In theory, to the compensated signal $s'(t) = \{s_1{}'(t), s_2{}'(t), \ldots, s_N{}'(t)\}$, the Doppler frequency modulation caused by the quadratic phase term will be eliminated; that means the long-time coherent integration can be realized to obtain higher SCR.

### 3.4. Target Discrimination

The extracted ROIs contain both moving targets and false alarms caused by sea clutter, so the target discrimination is needed to eliminate false alarms. For a target with a certain motion constraints, its spectrum $F(f_\tau)$ of the SIAF of a certain segment will have the form of $\sin c$ function that the peak is located at the center of Doppler frequency as shown in Equation (20), while that of sea clutter has stronger randomness, and the energy will be distributed in a wider range on the frequency domain. Therefore, the spectrum width of

the SIAF between the target and the sea clutter is obviously different, so the spectral width can be used for target discrimination. For the spectrum of the $K$th ROI, the 3 dB spectrum width can be calculated, denoted as $W_k$. For a ROI, if its $W_k$ satisfies inequality (25), it can be determined as the target.

$$W_k \leq \eta \cdot W_{3dB} \tag{25}$$

where $\eta$ is the threshold factor; $W_{3dB}$ is the 3 dB spectrum width of the $\sin c$ function; $W_{3dB} = 0.89 \cdot W_{Rayleigh}$ [33], where $W_{Rayleigh}$ is the Rayleigh bandwidth of the $\sin c$ function, which is related to the length of the time series. For a signal with a duration of $\tau$ s, its Rayleigh bandwidth is $W_{Rayleigh} = \frac{1}{\tau}$ Hz [34].

## 4. Experimental Verification

In order to verify the performance of the proposed long-time coherent integration based on segmented compensation, in this section, we organize the experiments based on simulated data and measured data, respectively. The simulation data are obtained by adding simulated low-SCR targets into the measured CSIR sea clutter. The measured data are collected by CSIR on the southwest coast of South Africa, which includes a moving boat.

### 4.1. Dim Targets Detection

Two simulated marine dim targets with very low SCR are added to the measured sea clutter to obtain the simulation data. The measured sea clutter is collected by CSIR on the southwest coast of South Africa in 2006, which is numbered as CFC17-001. The parameters of the radar observation are shown in Table 1.

**Table 1.** Parameters of radar observation.

| Parameters | Value |
|:---:|:---:|
| Frequency (GHz) | 9 |
| PRF (KHz) | 5 |
| Initial distance (m) | 3000.63 |
| Range resolution (m) | 15 |
| Grazing angle (deg) | 0.853–1.27 |
| Wind speed (m/s) | 7.97 |

For the simulated targets, the echo model of them uses the model which is introduced in Section 2.1. The motion of target 1 consists of two uniformly accelerated motions; the initial speed is 6 m/s; the accelerations of the two motions are $-2\,\text{m/s}^2$ and $1\,\text{m/s}^2$, respectively. It crosses the 21st and the 22nd range bins during the observation. The motion of target 2 includes three stages of uniformly accelerated motions with the initial speed of 12 m/s; the accelerations of the three motions are $4\,\text{m/s}^2$, $-3\,\text{m/s}^2$ and $2\,\text{m/s}^2$, respectively. It crosses the 51st and the 52nd range bins during the observation. The detailed motion parameters of the simulated targets are shown in Table 2.

**Table 2.** Motion parameters of simulated marine targets.

| Parameters | Target 1 | Target 2 |
|:---:|:---:|:---:|
| Initial distance (m) | 310 | 755 |
| Initial velocity (m/s) | 6 | 12 |
| Accelerations (m/s²) | −2, 1 | 4, −3, 2 |
| Duration (s) | 0.5, 0.5 | 0.3, 0.3, 0.4 |
| SCR (dB) | −15 | −17 |

The range–time image of simulated data is shown in Figure 2. The different colors represent different intensities of echoes. The simulated data consist of 96 range bins, and we select the 5000 pulses in 0–1 s for the experiment. Due to the low SCR, the targets are submerged in the sea clutter.

For the simulated data, after the five-layer wavelet packet decomposition is performed to each range bin, the frequency unit with high energy can be detected. The detection results are shown in Figure 3 where yellow represents the extracted high-energy units, while blue represents the lower-energy parts. It can be seen that the high-energy units are mainly concentrated in the low-frequency region.

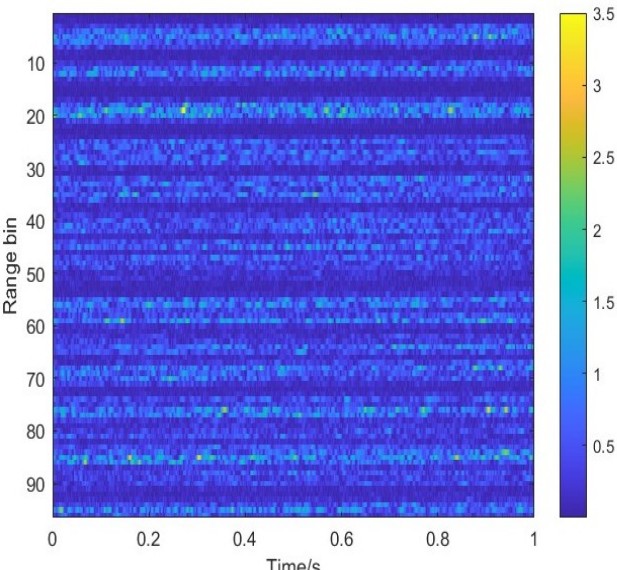

**Figure 2.** Range–time image of the simulation data.

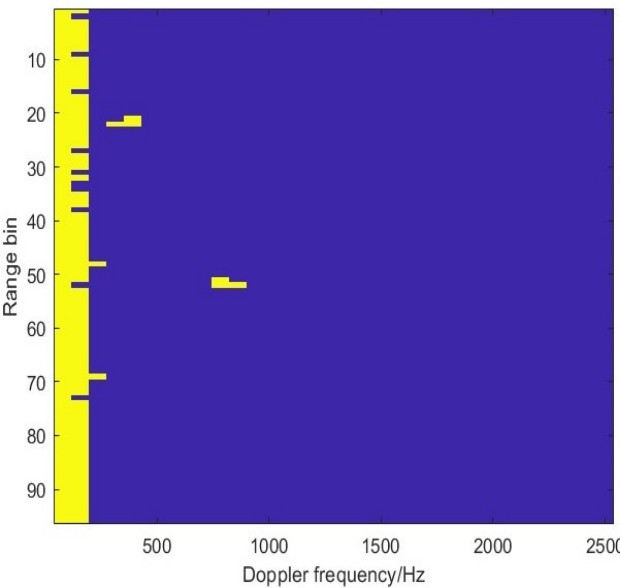

**Figure 3.** Detection result of the high–energy units.

After clutter suppression using Equation (18), the extracted ROIs are shown in Figure 4, which are labeled as 1, 2, 3, 4 according to the radial distance from radar, respectively. The range bins and Doppler frequency of the four ROIs are listed in Table 3. Among them, ROI 2 and ROI 4 are in the same Doppler frequency range; ROI 1 and ROI 3 cross different range bins, and ROI 2 and ROI 4 locate in the 48th range bin and the 66th range bin, respectively during the observation. For ROIs across different range bins, the time–frequency domain energy analysis described in Section 3.1 is used to select and extract the echo of the potential target.

**Table 3.** The location of the ROIs.

| ROI | Range Bin | Doppler Frequency/Hz |
| --- | --- | --- |
| ROI 1 | 21, 22 | 312.5~390.6 |
| ROI 2 | 48 | 156.3~234.4 |
| ROI 3 | 51, 52 | 781.3~859.4 |
| ROI 4 | 69 | 156.3~234.4 |

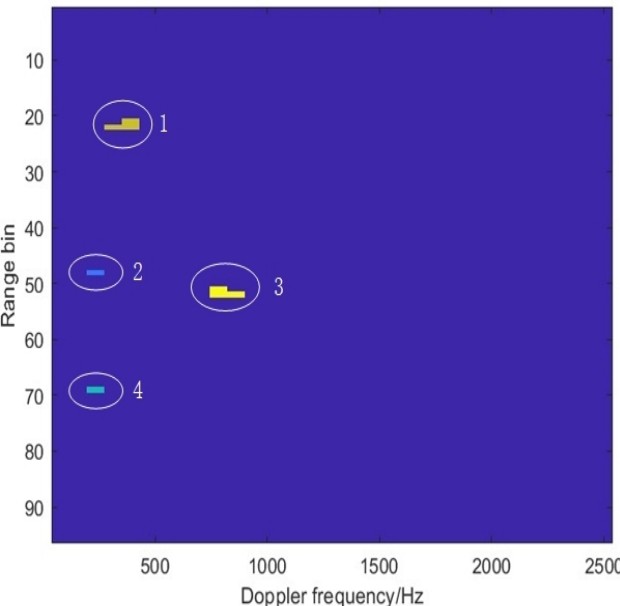

**Figure 4.** The extracted ROIs after clutter suppression.

Using the parameter estimation and motion segmentation described in Section 3.2, the parameters of the initial velocity and acceleration of each ROI are obtained. According to the parameters estimation, the motion of ROI 1 can be regarded as a combination of two stages of uniformly accelerated motion, from 0 to 0.5 s; the estimated initial velocity and acceleration are 5.92 m/s and $-1.84$ m/s$^2$, respectively; and in the range 0.5 to 1 s, the estimated initial velocity and acceleration are 5 m/s and 1.16 m/s$^2$, respectively. For ROI 2, its motion can be divided into three stages of uniformly accelerated motion, that is, 0 to 0.3 s, 0.3 to 0.4 s and 0.4 to 1 s; the estimated initial velocities are 2.92 m/s, 2.58 m/s and $-0.17$ m/s, respectively, the estimated acceleration are $-1.13$ m/s$^2$, $-27.5$ m/s$^2$, 4.87 m/s$^2$, respectively. The motion of ROI 3 can be segmented into three uniformly accelerated motions, that is, 0 to 0.3 s, 0.3 to 0.6 s and 0.6 to 1 s; the initial speed of the three motion segments is 12.17 m/s, 13.08 m/s, 12.17 m/s; and the acceleration is 3.03 m/s$^2$, $-3.03$ m/s$^2$, 2.9 m/s$^2$, respectively. The motion of ROI 4 can be divided into three stages of uniformly accelerated motion, from 0 to 0.6 s; the estimated initial velocity and acceleration are 2.75 m/s and $-3.05$ m/s$^2$, 0.6 to 0.8 s; its estimated initial velocity and acceleration are 0.92 m/s and 10.4 m/s$^2$; and from 0.8 to 1 s, its estimated initial velocity and acceleration are 3 m/s and $-5.85$ m/s$^2$, respectively.

Noise, data segmentation, clutter interference, etc., can affect the parameter estimation. The mean square error (MSE) is used to measure the accuracy of parameter estimation of the targets, which is calculated as $MSE = \frac{1}{n} \sum_{i=1}^{n} \left( Y_i - \hat{Y}_i \right)^2$, where $\hat{Y}_i$ is the estimation of $Y_i$. The MSE of velocity estimation and acceleration estimation of target 1 are $6.2 \times 10^{-3}$ m/s and $2.56 \times 10^{-2}$ m/s$^2$, respectively; The MSE of velocity estimation and acceleration estimation of target 2 are $4.9 \times 10^{-2}$ m/s and 0.58 m/s$^2$, respectively. It can be seen that the estimation result is relatively accurate. It should be noted that the excessive parameter estimation errors can lead to significant phase mismatch, which will cause

Doppler spectrum expansion cannot be effectively eliminated. Because the Doppler frequency of sea clutter is affected by many environmental factors, it has strong temporal variability, which is difficult to describe its mechanism by specific model. Therefore, according to the parameter estimation and segmentation results of ROI 2 and ROI 4 shown in Figure 5b,d, the changes of velocity and acceleration have strong randomness and are significantly different from those of marine targets with specific motion models.

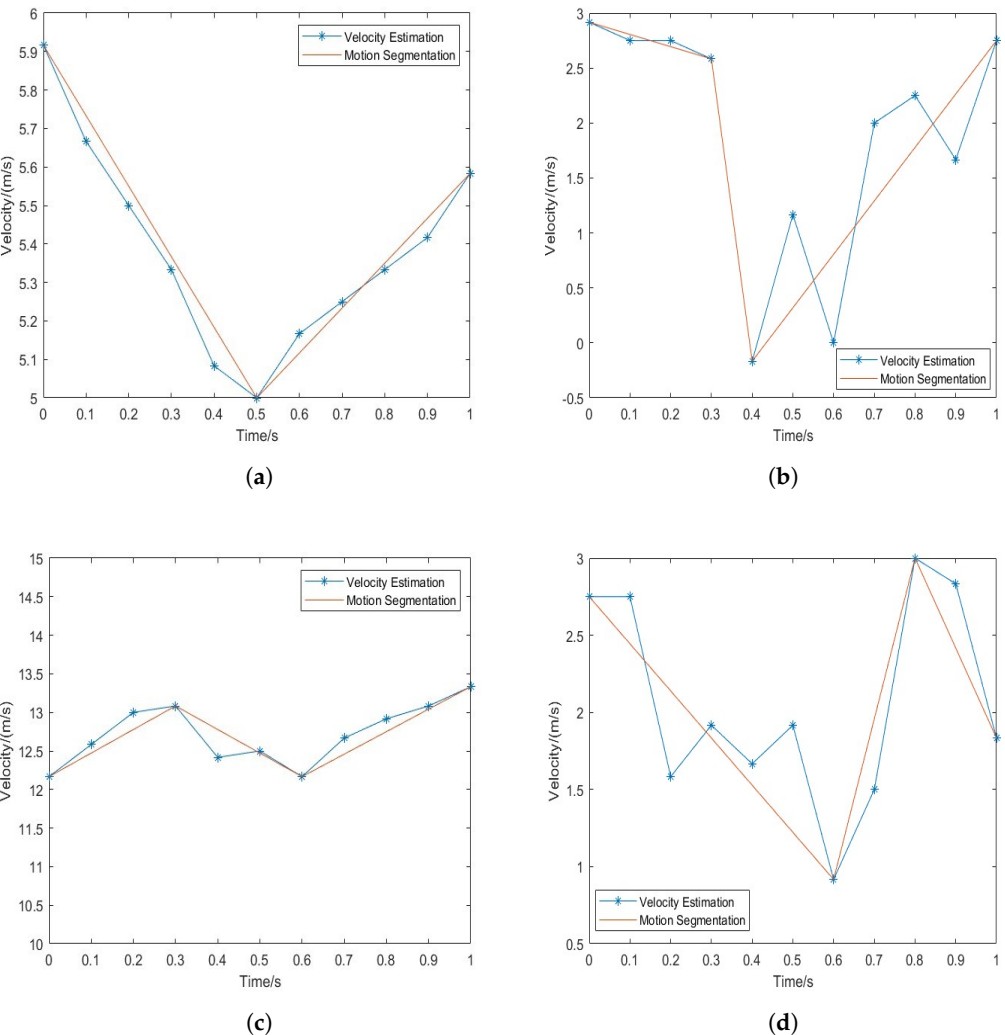

**Figure 5.** Parameter estimation and motion segmentation results of 4 ROIs of simulation data, where (**a**–**d**) correspond to ROI 1-ROI 4, respectively.

According to the estimation of motion parameters and the segmentation, the extracted echo of four ROIs can be compensated in each segment, respectively. Then the long-time coherent integration can be performed to obtain high SCR. Taking ROI 1 as an example, the segmented compensation integration results are shown in Figure 6a. MTD is a typical target detection method, which is based on coherent integration; as a comparison, the integration effect of MTD of ROI 1 is also given in Figure 6b. As can be seen from Figure 6, the proposed segmented coherent integration can well compensate the Doppler frequency modulation caused by the variable acceleration movement of the marine target, so that the target can be accumulated in a small range of frequency domain. To finely describe the integration results, the SCR is calculated as the ratio between the maximum power of the target and the average power of sea clutter. The SCR after segmented compensation integration is 19.7 dB, which is significantly higher than that of the coherent

integration of MTD, which is 14.7 dB. In Figure 6, both the sea clutter and the target correspond to peaks, to distinguish them clear, the target is circled in red.

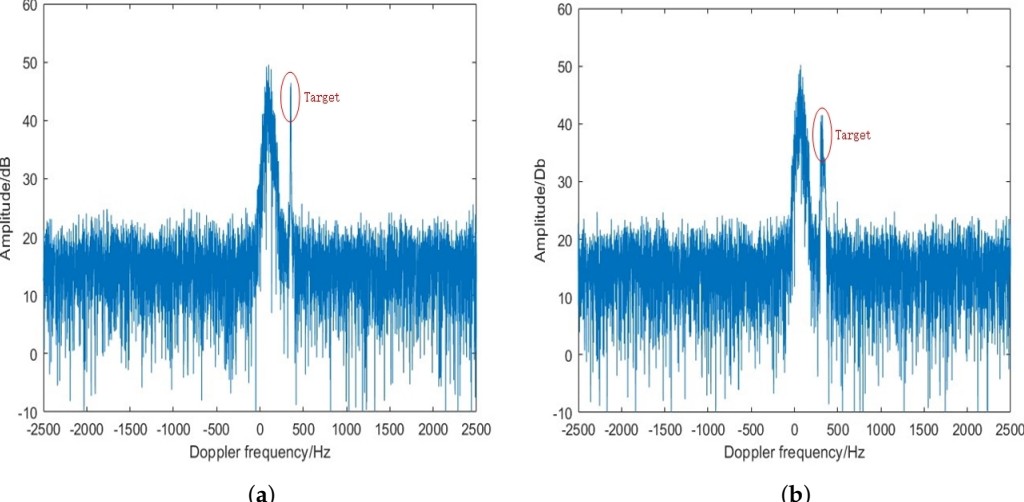

(**a**)  (**b**)

**Figure 6.** Comparison of MTD and proposed method in ROI 1. (**a**) Integration result of proposed method. (**b**) Integration result of MTD.

The detection of ROIs is performed under low SCR, and there will be false alarms caused by sea clutter, so it is necessary to discriminate the extracted ROIs. For each ROI, the 3 dB spectral width of SIAF is calculated. Based on the motion segmentation of four ROIs, the segment with the longest duration of each ROI is selected for target discrimination. The selected segments of ROI 1-ROI 4 are 0 s to 0.5 s, 0.4 s to 1 s, 0.6 s to 1 s and 0 s to 0.6 s, respectively. For the selected segments, the standard 3 dB spectral width of SIAF are about 1.78 Hz, 1.48 Hz, 2.225 Hz and 1.48 Hz, respectively, taking the coefficient $\eta = 1.5$, the spectral width thresholds are calculated. The target discrimination results are shown in Table 4, where ROI 1 and 3 are the areas where the targets exist, which is consistent with the setting of the simulation. By using long-time coherent integration, the very low SCR targets are detected successfully.

Figure 7 shows the spectrum of the SIAF of the four ROIs, where the red line denotes the 3 dB attenuation of the maximum value. As can be seen from Figure 7, due to the irregular motion of the sea clutter, the 3 dB spectral width of the SIAF is wider than that of the marine targets with clear motion constraints.

**Table 4.** Target discrimination of ROIs.

| ROI | 3 dB Band Width/Hz | Threshold/Hz | Discrimination |
|-----|--------------------|--------------|----------------|
| ROI 1 | 2 | 2.67 | T |
| ROI 2 | 70.8 | 2.23 | F |
| ROI 3 | 2.5 | 3.34 | T |
| ROI 4 | 64.3 | 2.23 | F |

In this experiment, the proposed long-time coherent integration based on segmented compensation achieves the energy integration and target detection of two targets with complex motion under very low SCR, and effectively removes the false alarms, indicating that the proposed method has the ability of marine dim target detection with complex motion.

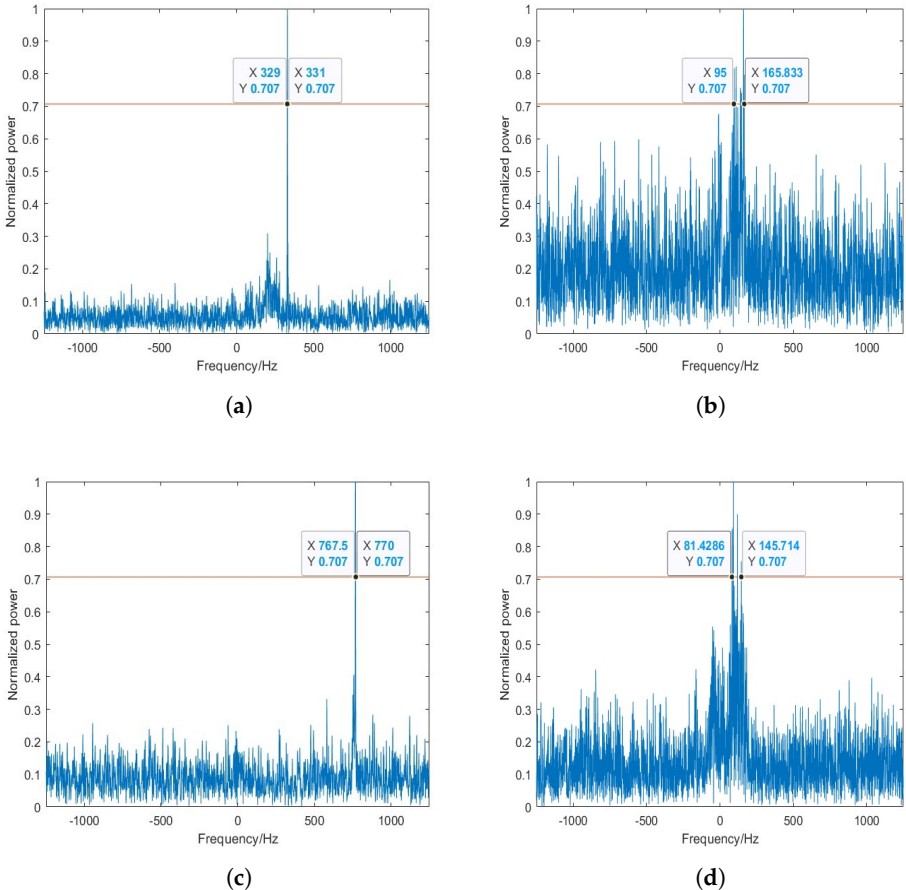

**Figure 7.** The spectrum of symmetric instantaneous autocorrelation function of the 4 ROIs in simulated data, where (**a**–**d**) correspond to the spectrum of symmetric instantaneous autocorrelation function of ROI 1-ROI 4, respectively.

*4.2. Detection Performance Simulation*

Target detection from sea clutter is affected by many factors in order to verify the performance of the proposed method. Monte Carlo simulation under different SCR is used. The sea clutter can be simulated using K-distribution model, in which a variable acceleration target is randomly inserted. The initial velocity of the target takes a random value in the range of 0~10 m/s, and the acceleration takes a random value in the range of $-5$~5 m/s$^2$. Each target includes two stages of uniformly accelerated motions.

In the experiments, to compare the detection performance, four methods of coherent integration: segmented compensation integration, RFrFT, RFT and MTD, are used to realize the coherent integration; CFAR detection was performed to detect the targets. The false alarm rate of CFAR detection was set to $10^{-4}$, and the times of the Monte Carlo experiments were set to $10^4$. The detection probability under each SCR was calculated, and the ROC curve is shown in Figure 8. According to the simulation results, under the same SCR, the detection probability of the proposed method is higher than those of RFrFT, RFT and MTD. To obtain the same detection probability, the required SCR of the proposed method is about 4.5 dB less than that of the RFrFT, and about 6 dB less than that of the RFT. The simulation results show that the proposed method can effectively improve the dim target detection performance under heavy sea clutter, and the detection probability is higher than that of RFrFT, RFT and MTD under the same conditions, especially to lower SCR.

If there are multiple targets in the actual scene and relatively close, then the threshold of the CA-CFAR detector will be affected by the neighboring targets, and the detection performance will decrease.

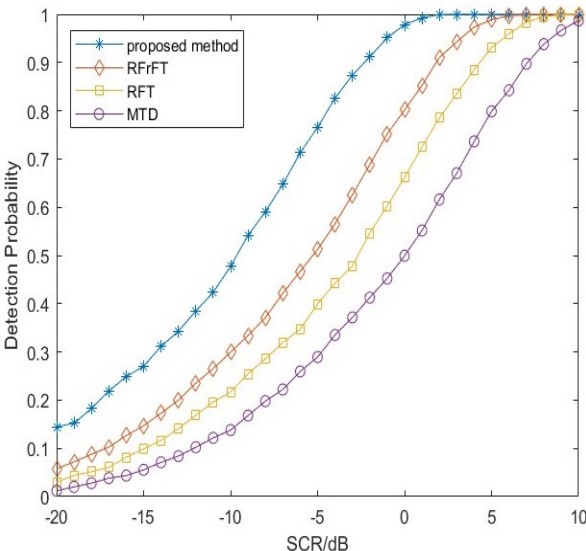

**Figure 8.** Simulation results of Pd-SCR.

### 4.3. Target Detection Based on Measured Data

In order to verify the detection effect of the proposed method on the actual scene, the measured CSIR data TFC17-006, which were also collected from the southwest coastline of South Africa in 2006, were used for the experiment. The radar was operating in X-band, with a range resolution of 15m and a pulse repetition frequency of 5000 Hz. During the radar observation, the wind speed was 11.96 m/s and the grazing angle was from 0.501 to 0.560 degree. Like Ref. [26], we also selected 5000 pulses between 80.5 to 81.5 s for the experiment. A boat, as the target, was located in the 23rd and 24th range bins in the data. The range–time image of the measured data is shown in Figure 9.

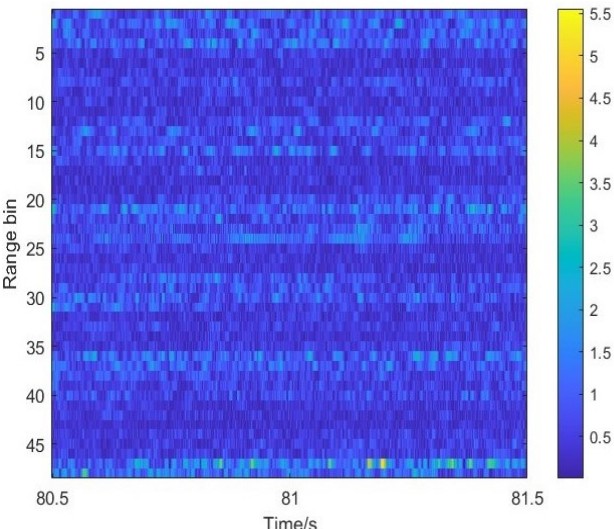

**Figure 9.** Range–time image of the measured CSIR data.

Same as the Ref. [26], four ROIs can be extracted from this experimental scenario, as shown in Figure 10. The four ROIs are labeled as 1, 2, 3 and 4, respectively, according to the radial distance from radar. Among them, ROI 1 is located in the 17th range bin; ROI 2 is distributed in the 23rd and 24th range bins; ROI 3 and ROI 4 are located in the 28th range bin and 44th range bin, respectively. The Doppler frequency of the four ROIs range from 312.5 Hz to 468.75 Hz. For the extracted potential targets, the method described in Section 3.2 is used for motion parameter estimation and segmentation.

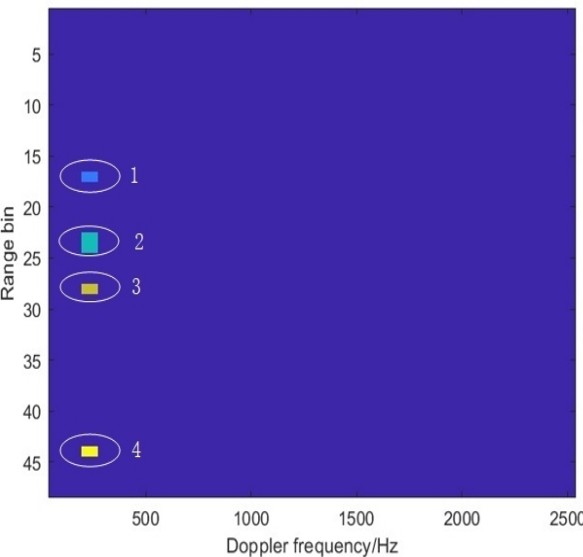

**Figure 10.** The extracted ROIs from the measured data.

To the extracted ROIs, the velocity estimation and motion segmentation results are shown in Figure 11. Among them, ROI 2 is caused by a real boat. Just like Figure 11b, the movement of the boat is divided into 3 segments. ROI 1, ROI 3 and ROI 4 are false alarms caused by sea clutter, which shows the irregular movement. The parameters of velocity and acceleration in each segment can be estimated to construct the phase compensation factor.

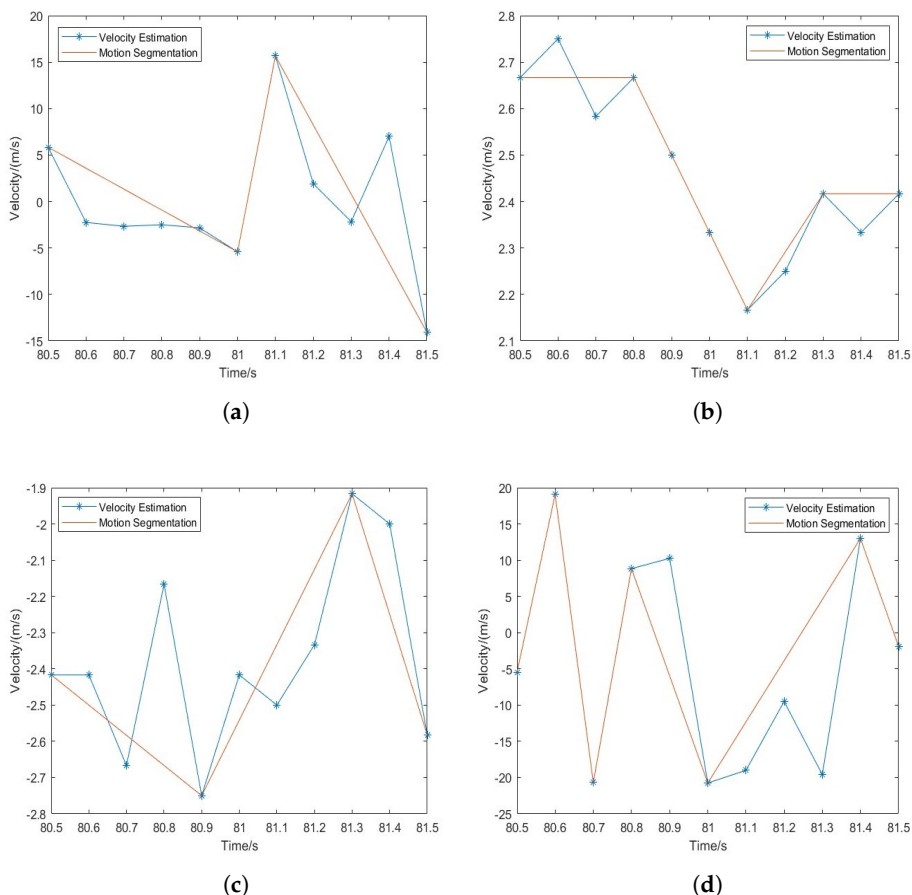

**Figure 11.** Parameter estimation and motion segmentation results of 4 ROIs of measured data, where (**a**–**d**) correspond to the results of ROI 1-ROI 4, respectively.

Figure 12 shows the spectrum of the SIAF of the 4 ROIs; the red line in the Figure 12 shows the 3 dB attenuation of the maximum value. The selected motion segments of ROI 1-ROI 4 are 80.5 s to 81 s, 80.8 s to 81.1 s, 80.9 s to 81.3 s and 81 s to 81.4 s, respectively. The target discrimination of the four ROIs is performed based on the 3 dB spectral width of the SIAF of the selected segments. The 3 dB spectrum width of the four ROIs are 22 Hz, 3.33 Hz, 68.75 Hz and 50 Hz, respectively. Taking the coefficient $\eta = 1.5$, the spectral width thresholds are 2.67 Hz, 4.45 Hz, 3.34 Hz, 3.34 Hz, respectively. For ROI 2, its 3 dB spectral width of the SIAF is less than the spectral width threshold, which is discriminated as the target. ROI 1, ROI 3 and ROI 4 are identified as false alarms caused by sea clutter. The result of target discrimination is consistent with the ground truth.

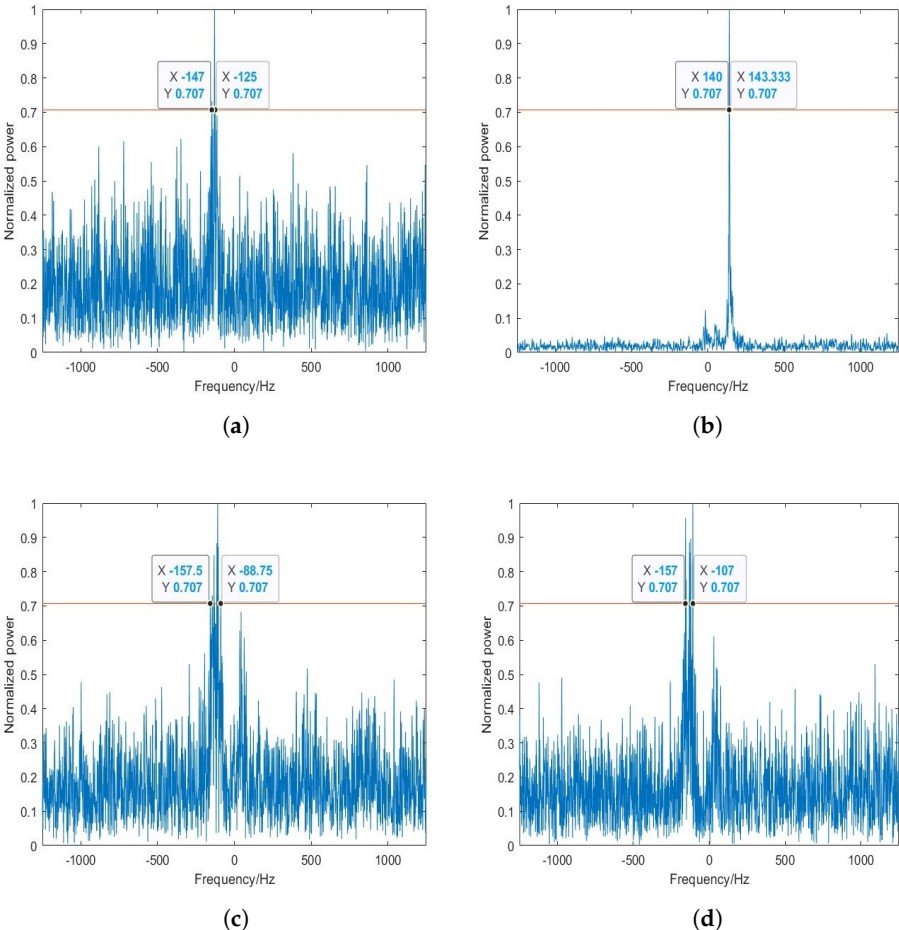

**Figure 12.** The spectrum of symmetric instantaneous autocorrelation function of the 4 ROIs in measured data, where (**a**–**d**) correspond to the spectrum of symmetric instantaneous autocorrelation function of ROI 1-ROI 4, respectively.

After segmented compensation, the integration results are shown in Figure 13. In Figure 13, the integrated energy of target is higher than the energy of sea clutter. Experiment results indicate that the long-time coherent integration based on segmented compensation can effectively compensate the Doppler frequency modulation caused by the complex motion of the marine target, and can realize the efficient integration of target energy. As a comparison, the integration effect of MTD in the 24th range bin and that of RFrFT in ROI 2 are also given in Figure 14a,b. As can be seen from Figures 13 and 14, the proposed method can significantly improve the SCR after coherent integration.

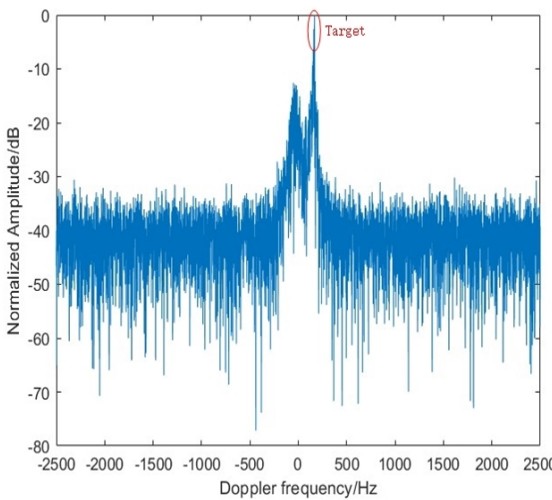

**Figure 13.** Integration result of proposed method.

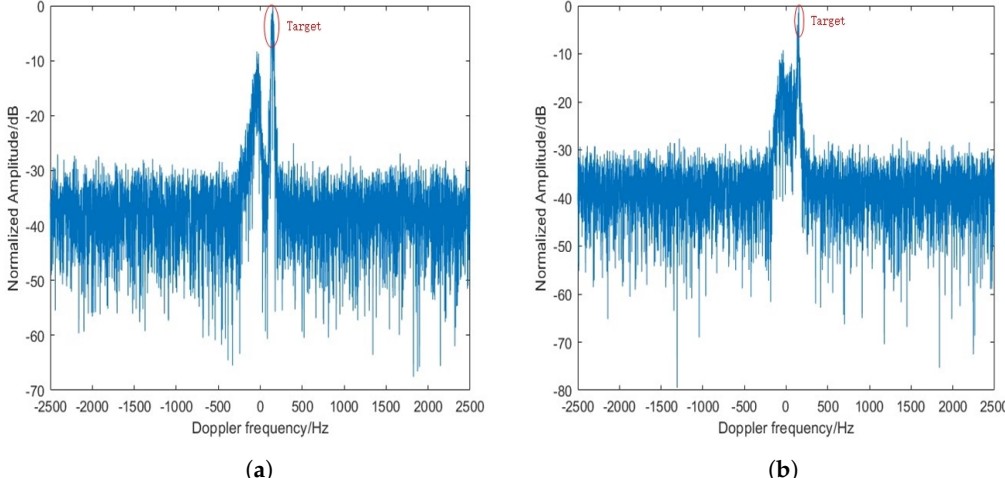

(**a**)  (**b**)

**Figure 14.** Integration results of MTD and RFrFT. (**a**) Integration result of MTD in 24th range bin. (**b**) Integration result of RFrFT in ROI 2.

In order to show the integration effect of the proposed method on the boat in measured data more clearly, Table 5 lists the integration effect of MTD, RFrFT and the proposed method. It can be seen that the proposed method can obtain the highest SCR, which is about 4.5 dB higher than that of the representative RFrFT method. The experimental results of the measured data verify that the proposed method can effectively improve the SCR of complex moving target, and is an effective dim target detection method.

**Table 5.** The SCR of different integration methods.

| Methods | SCR/dB |
|---|---|
| MTD of 23rd range bin | 25.9 |
| MTD of 24th range bin | 26.3 |
| RFrFT | 28.5 |
| Proposed method | 33.0 |

## 5. Conclusions

In this paper, a long-time coherent integration method based on segmented compensation is proposed for the maneuvering target detection from sea clutter. The proposed method

models the complex motion of the maritime target as a multi-stage uniformly accelerated motion with different parameters. By estimating the moving parameters of each stage, the Doppler frequency modulation caused by the acceleration will be eliminated. And finally, the long-time coherent integration is realized, which will increase the detection performance of dim targets with complex movement.

Based on multi-stage motion modeling, the quadratic phase of the complex moving target can be compensated through parameter estimation in each stage. The processing of segmented compensation provides a new method to realize long-time coherent integration to a complex moving target.

Maritime target detection is affected by various factors, especially at low SCR. To eliminate false alarms, a novel threshold based on the 3 dB bandwidth of the spectrum of symmetric instantaneous autocorrelation function is constructed for target discrimination. The threshold is effective for determining whether an ROI is a target or sea clutter.

The target detection experiment based on simulated data show that the proposed method can extract ROIs, estimate moving parameters and detect targets successfully from sea clutter under very low SCR of $-15$ dB and $-17$ dB. At the same time, it can obtain the best detection performance than the representative detection methods, such as MTD, RFT and RFrFT. Under the SCR of $-5$ dB, the detection probability of the proposed method is about 0.3 higher than that of RFrFT, 0.4 higher than that of RFT and 0.5 higher than that of MTD. The result indicates that the proposed method has good detection performance for weak and small targets with very low SCR. The experiment based on the measured CSIR data show that the proposed method can effectively integrate target energy to obtain higher SCR than the RFrFT of 4.5 dB and detect the real boat from sea clutter. The experimental results show that the proposed long-time coherent integration method based on segmented compensation can effectively integrate complex moving targets' energy to detect them from sea clutter.

**Author Contributions:** Conceptualization, Z.Z. and W.W. (Wenguang Wang); methodology, Z.Z. and Y.Z.; software, Z.Z. and Y.Z.; validation, W.W. (Wenguang Wang), B.L. and W.W. (Wei Wu); formal analysis, Z.Z. and Y.Z.; investigation, Z.Z.; resources, W.W. (Wenguang Wang); data curation, Z.Z. and Y.Z.; writing—original draft preparation, Z.Z.; writing—review and editing, W.W. (Wenguang Wang) and B.L.; visualization, Z.Z.; supervision, W.W. (Wenguang Wang); project administration, W.W. (Wenguang Wang); funding acquisition, W.W. (Wei Wu). All authors have read and agreed to the published version of the manuscript.

**Funding:** This research was funded by the National Natural Science Foundation of China, Grant No. 62073334 and Grant No. 61771028.

**Data Availability Statement:** Not applicable.

**Conflicts of Interest:** The authors declare no conflict of interest.

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
