# Peer review of "Long-Time Coherent Integration for Marine Targets Based on Segmented Compensation"

_remotesensing, doi:10.3390/rs15184530_

Round 1

Reviewer 1 Report

An interesting and apt research for real time applications in the current scenario.

Well presented article.

Author Response

Thanks for your reviewing.

Reviewer 2 Report

In this paper, it is proposed that the long-term coherence integration of Marine targets based on segmentation compensation can finally realize the detection and recognition of small and small targets under low SNR by detecting the region of interest, parameter estimation and segmentation, phase compensation and long-term coherence accumulation. The proposed method can achieve better detection performance for complex moving targets under low SNR, but there are the following problems.

1.     After the results of parameter estimation and segmentation of ROI1-ROI4 shown in Figure 5, why is there no spectral diagram of symmetric instantaneous autocorrelation function of 4 detection regions?

2.     Does the number of detection targets in the comparison simulation shown in Figure 7 in Section 4.2 affect the detection performance of the proposed method and comparison experiment?

3.     Does the error size of parameter estimation and segmentation have a great influence on the subsequent phase compensation and long-term coherent accumulation?

4.     For the four ROIs detected in Figure 4, it is recommended to tabulate the data for each ROI.

The quality of English should be improved further.

Author Response

Comment 1

After the results of parameter estimation and segmentation of ROI1-ROI4 shown in Figure 5, why is there no spectral diagram of symmetric instantaneous autocorrelation function of 4 detection regions?

Response

We have added the spectral diagram of symmetric instantaneous autocorrelation function of the 4 ROIs (Figure 7) in the Section 4.1.

Comment 2

Does the number of detection targets in the comparison simulation shown in Figure 7 in Section 4.2 affect the detection performance of the proposed method and comparison experiment?

Response

In the experiment,the CA-CFAR is used as the detector. If the targets are relatively close, multi-target will reduce the detection performance. We have explained the impact of multiple targets on the detection performance in the Section 4.2 (p.14 l.413-l.415).

Comment 3

Does the error size of parameter estimation and segmentation have a great influence on the subsequent phase compensation and long-term coherent accumulation?

Response

Excessive parameter estimation errors can lead to significant phase mismatch, which will cause Doppler spectrum expansion cannot be completely eliminated, so errors in parameter estimation and motion segmentation will have an impact on the long-time integration. We have added the impact of parameter estimation errors on long-time coherent integration in Section 4.1 (p.12 l.356-l.358).

Comment 4

For the four ROIs detected in Figure 4, it is recommended to tabulate the data for each ROI.

Response

We have listed the range bins and the Doppler frequency of the ROIs in Table 3.

Reviewer 3 Report

1. Author mention the add problem clearly , lack of literature review ?

2.figure 2 need more detail explanation.

3.  Simulation parameters need more detail each parameters should be clearly.

4. Most of result need detail explanation author just insert the results more of details are missing.

5. Flow charts need add the other steps, algorithm step and detail are missing.most of equations should be clearly explain .

6. Conclusion should be expand also author need to compare their recently methods

Many typos errors and mistake 

Author Response

Comment 1

Author mention the add problem clearly , lack of literature review?

Response

We have added the literature review in the section of Introduction (p.2 l.43-l.45 l.62-l.66).

Comment 2

Figure 2 need more detail explanation.

Response

The detailed information of Figure 2 is added in Section 4.1 (p.10 l.319-l.321). Moreover, some observation parameters of the radar are added in Table 1.

Comment 3

Simulation parameters need more detail each parameters should be clearly.

Response

In the Section 4.1, we explained the simulation parameters in detail, which are shown in p.9 l.312-l.316. At the same time, some observation parameters of the radar are listed in Table 1 to help understand the simulation settings.

Comment 4

Most of result need detail explanation author just insert the results more of details are missing.

Response

According to the suggestion, in the Section 4.3, we have explained the experiments results in detail (p.15 l.427-l.430).

Comment 5

Flow charts need add the other steps, algorithm step and detail are missing.most of equations should be clearly explain.

Response

We added explanation of the algorithm process, especially for those that include multiple modules in Section 3.1 (p.6 l.199 l.200) and Section 3.2 (p.7 l.241-l.244).

Comment 6

Conclusion should be expand also author need to compare their recently methods.

Response

Recently, some researchers focus on using the higher order motion to model the motion of the maneuvering target especially to the near space targets. But in our opinion, for the marine targets, multi-stage uniformly accelerated motion is enough to model them. So, we just compare our method with several representative methods, including MTD and RFrFT. According to the suggestion, we expand the conclusion (p.18 l.481-l.484).

Reviewer 4 Report

 In this article, a new modeling method that decomposes the complex motion of the target into the combination of multiple uniformly accelerated motions to achieve a simplified description. Besides, a new target discrimination based on spectrum width is given to eliminate the false alarms. Generally, a novel detection method for maneuvering weak targets is provided in this paper.

The article is organized well. However, the following points should be addressed:

1)  The idea of using the integrated signal bandwidth to discriminate targets is very interesting. In Table 2, the 3dB bandwidth of ROI1 andROI3 is 2Hz and 3Hz, respectively. In fact, it is difficult to obtain such high frequency resolution, especially in very low SCR. The authors can further explain which factors are related to this bandwidth.

2) In the caption of Figure 6, the order of Figures a and b is reversed.

3) In the sentence “The extracted segments are connected successively to complete the distance migration correction of the echo.” (p.7, l.230), the concept of "distance migration correction" is not very exact.

4) In Section 1, “which need to search the acceleration of the moving target” (p.2, l.53) would be “which needs to search the acceleration of the moving target”; “method in reference 23” (p.2, l.63) would be “the method in reference 23”.

5) In the conclusions, “the propose method” (p.17, l.454) should be “the proposed method”.

The English language is fine. 

Author Response

Comment 1

The idea of using the integrated signal bandwidth to discriminate targets is very interesting. In Table 2, the 3dB bandwidth of ROI1 andROI3 is 2Hz and 3Hz, respectively. In fact, it is difficult to obtain such high frequency resolution, especially in very low SCR. The authors can further explain which factors are related to this bandwidth.

Response

The frequency resolution is related to the number of integration pulses. Under a certain PRF, the longer the coherent integration time, the higher the frequency resolution.

Comment 2

In the caption of Figure 6, the order of Figures a and b is reversed.

Response

We have reversed the caption of Figure 6 (a) and Figure 6 (b).

Comment 3

In the sentence “The extracted segments are connected successively to complete the distance migration correction of the echo.” (p.7, l.230), the concept of "distance migration correction" is not very exact.

Response

According to the suggestion, we have deleted the inaccuracy concept and described it more appropriately (p.7 l.237-l.239).

Comment 4

In Section 1, “which need to search the acceleration of the moving target” (p.2, l.53) would be “which needs to search the acceleration of the moving target”; “method in reference 23” (p.2, l.63) would be “the method in reference 23”.

Response

Accepted and revised.

Comment 5

In the conclusions, “the propose method” (p.17, l.454) should be “the proposed method”.

Response

Accepted and revised.

Round 2

Reviewer 3 Report

All comments are well addresses